# Emerging Advancements in Piezoelectric Nanomaterials for Dynamic Tumor Therapy

**DOI:** 10.3390/molecules28073170

**Published:** 2023-04-02

**Authors:** Qian Yu, Wenhui Shi, Shun Li, Hong Liu, Jianming Zhang

**Affiliations:** 1School of Life Science, Jiangsu University, Zhenjiang 212013, China; 2School of Chemistry and Chemical Engineering, Jiangsu University, Zhenjiang 212013, China

**Keywords:** piezoelectric, nanomaterial, tumor therapy, reactive oxygen species, ultrasound

## Abstract

Cancer is one of the deadliest diseases, having spurred researchers to explore effective therapeutic strategies for several centuries. Although efficacious, conventional chemotherapy usually introduces various side effects, such as cytotoxicity or multi−drug resistance. In recent decades, nanomaterials, possessing unique physical and chemical properties, have been used for the treatment of a wide range of cancers. Dynamic therapies, which can kill target cells using reactive oxygen species (ROS), are promising for tumor treatment, as they overcome the drawbacks of chemotherapy methods. Piezoelectric nanomaterials, featuring a unique property to convert ultrasound vibration energy into electrical energy, have also attracted increasing attention in biomedical research, as the piezoelectric effect can drive chemical reactions to generate ROS, leading to the newly emerging technique of ultrasound−driven tumor therapy. Piezoelectric materials are expected to bring a better solution for efficient and safe cancer treatment, as well as patient pain relief. In this review article, we highlight the most recent achievements of piezoelectric biomaterials for tumor therapy, including the mechanism of piezoelectric catalysis, conventional piezoelectric materials, modified piezoelectric materials and multifunctional piezoelectric materials for tumor treatment.

## 1. Introduction

Cancerous tumors, as one of the deadliest diseases and the second leading cause of human death, can grow uncontrollably and spread in the body, which inhibits the division or growth of normal cells and causes their death [1]. Numerous strategies have been developed to treat solid tumors, including surgery [2], radiotherapy [3] and chemotherapy [4]. Although effective in suppressing the multiplication of cancerous cells, clinical evidence reveals that medicines also cause severe side effects in patients, such as appetite loss, vomiting and hair loss, and harm normal cells. Therefore, it is crucial to explore alternative medications with low toxicity and minimal side effects for the treatment of malignancies [5]. In recent years, novel strategies, using nanomaterials (NMs) as the active center and external fields as the driven energy, have been innovated for the treatment of cancers, such as photothermal therapy (PTT), photodynamic therapy (PDT), sonodynamic therapy (SDT) [6] and chemodynamic therapy (CDT) [7], where the implanted NMs are excited by the exogenous field to produce reactive oxygen species (ROS) to destroy cancer cells and to effectively inhibit their regrowth [8]. Moreover, the NMs are highly biocompatible to minimize the harm to normal cells. 

NMs are novel functional materials with unique physical and chemical characteristics, outperforming their bulk counterparts. Due to their small size, NMs may pass through biological barriers to interact with key substances inside the cell, making them an ideal platform for biomedicine [9,10]. Among the various types of NMs, piezoelectric nanostructures feature the conversion of mechanical energy to electrical energy via the piezoelectric effect. Piezoelectric materials are highly sensitive to mechanical vibration; even a weak force, such as water flow, breath and muscle movement, can generate micro electrical charges. Recently, piezoelectric NMs have been used as catalysts to convert mechanical micro−energy, e.g., ultrasonication, into electric energy to drive chemical reactions, including water splitting, depollution, organic polymerization and so on [11,12,13,14,15]. In this so−called piezoelectric catalysis, electrons and holes can be continuously separated upon mechanical force stimulation to form highly active radicals.

Upon the conversion of mechanical energy into electrical energy, piezoelectric materials have been investigated in bio−research of cell stimulation, antibacterial, tissue engineering as well as sensing. Recently, the concept of piezoelectric catalysis has also been introduced for tumor treatment as a rapidly emerging technique [16]. In particular, due to the ultra−high penetration of ultrasonic waves in organisms, the non−invasive nature of ultrasound and reduced harm to healthy tissues, a safe and effective treatment for deep malignant tissues becomes feasible using ultrasound and piezoelectric biomaterial [17]. The piezoelectric catalysis, taking place in the tumor, produces highly concentrated ROS, including ^1^O_2_, •O_2_^−^ and •OH, which are able to trigger programmed cell death, similarly to the dynamic therapies [18,19], as illustrated in Figure 1; this is known as piezoelectric dynamic therapy (PEDT). Compared to the photo−energy−converting NM−based PPT and PDT process, piezoelectric NM−based PEDT demonstrates the unique advantages of less damage to healthy tissue, non−invasive process and effectiveness to cure deep−seated tumors. 

In this review, we highlight the recent advancements in piezoelectric NM−based dynamic therapy for cancer treatment. We start with a brief of the mechanism of piezoelectric catalysis and then move to the introduction of the main piezoelectric bio−NMs followed by showing their representative applications in cancer therapy. This review is concluded by proposing some challenges in this research domain and several interesting research directions in the future. We hope it can serve as a valuable reference toward developing more safe and efficient biomedicines and motivate more extensive bio−application with these NMs.

## 2. Mechanism of Piezoelectric Catalysis

In general, piezoelectric materials are mainly composed of noncentrosymmetric crystals. This structural asymmetry originates from the anisotropic arrangement of oppositely charged ions in the materials. Under the action of applied stress, the atoms are displaced from their original lattice position, resulting in a net electrical charge over the whole crystal, which can be uniformly oriented in a microdomain with a mutual counterbalance of the corresponding charges. The electric dipoles in the domain can be aligned by exerting a mechanical stress to the material, which results in a separation/redistribution of the oppositely charged ions to the two sides of the material, eventually forming a built−in electric field and inducing band bending of the semiconductor [20], as presented in Figure 2. When the surface charges actively promote or obstruct the electrochemical reactions, taking place at the interface between the material and the solution, this is referred to as piezoelectric catalysis or piezocatalysis [21,22].

Piezoelectric catalysis is therefore also known as a catalytic reaction driven by the use of piezoelectric stress or strain [23]. In cancer treatment, the ultrasonic wave as the mechanical force, spreading in the body, can distort the surface of a piezoelectric material implanted in the tissue, thus initiating the piezoelectric effect by creating an internal electric field, which drives the separation of positive/negative charges to the opposite directions of the material to participate in surface reactions to generate ROS [24]. This simple yet effective concept has led to a rapidly emerging application of PEDT in cancer treatment.

## 3. Main Piezoelectric NMs for PEDT

Superior bio−safety and stability of implanted materials are crucial for biomedicine application, which greatly limits the type of piezoelectric materials. In general, prior to the implantation of a material into an organism, the cytotoxicity of the NMs must be systematically studied to obtain the information on the possible negative impacts on the cells, which are typically present in living creatures [25]. The most simple and visible test is to assess cell viability as a function of material concentration, which can serve as a pre−requisite for potential animal research [26]. The cytotoxicity of the piezoelectric effect has not been reported, since the piezoelectricity generated by NMs is rather weak and only takes place in a confined region of the material surface. In fact, some components in human bones, i.e., hydroxyapatite (Ca_10_(PO_4_)_6_(OH)_2_), also display low piezoelectric properties. When the bones are stressed, surface charges can be produced after the piezoelectric effect [27,28], whereas piezoelectricity is too weak to be transmitted for ROS formation.

Barium titanate (BaTiO_3_, BTO) and zinc oxide (ZnO) are the most available, safe and classic inorganic piezoelectric materials. BTO, ZnO and their hybrids, despite their limited piezoelectric properties, have been utilized for piezoelectric catalysis and PEDT, owing to their high bio−safety. Lead−based piezoelectric materials, such as PbTiO_3_, although possess high piezoelectric property, are toxic and thus rarely employed for biological applications. Organic piezoelectric materials, such as polyvinylidene fluoride (PVDF), show good bio−compatibility and flexibility, but they only show minor piezoelectric properties, which usually must be enhanced by additional polarization treatment with a strong electric field. 

ZnO nanoparticles are a Generally Recognized as Safe (GRAS) material by the U.S. Food and Drug Administration (FDA) [28]. ZnO is one of the most cost−effective and widely used inorganic materials, from research to industry, due to its low cost, multiple dielectric properties and safety [29]. Hexagonal wurtzite ZnO is one of the most broadly studied piezoelectric materials, whose spontaneous polarization usually forms along its longitudinal direction (c−axis) (Figure 3a) [30]. The centers of the positive and negative ions overlap in a positively tetrahedral coordination between Zn^2+^ and O_2_^−^. When force is applied at the tetrahedron’s apex, the centers of the cation and the anion are moved apart from one another, creating a dipole moment. A piezoelectric field is then created by the constructive addition of all the dipole moments produced by the crystal units, reducing the crystal macroscopic potential in the direction of the strain, which is also known as the piezoelectric potential, as illustrated in Figure 3b [30]. 

Additionally, researchers have also discovered that Zn^2+^, one of the trace elements needed by the human body, plays a pivotal role in tumor treatment. Zn^2+^ is capable of regulating a variety of bodily functions that can destroy some cancer cells and is readily absorbed and metabolized by cells [31]. For instance, a recent report showed that ZnO nanoparticles were harmful to cancer cells but not to healthy cells. Wang et al. [32] studied the cytotoxicity of one−dimensional (1D) ZnO nanowires and discovered that they were significantly more lethal than their equivalent spherical nanoparticles. Studies on the cytotoxicity of nanoflowers and nanorods showed that ZnO can inhibit the proliferation of cancer cells by overcoming the drug resistance of tumor cells [33,34]; ZnO has also shown to have a good targeting effect on a wide range of cancer cells [35]. This advantage has a good curative effect on patients with multiple types of cancers, in addition to their piezoelectric property. 

BTO is known as the only piezoelectric compound amongst the chalcogenide materials with ferroelectric capabilities. Under strain, the displacement of Ti atoms from the central position of the TiO_6_ octahedron induces the spontaneous polarization of BTO (p_0_), as shown in Figure 4a [36]. The piezoelectric response can be characterized by the hysteresis loop and butterfly curve by piezo−response force microscopy (PFM) (Figure 4b), and the piezoelectric field distribution can be simulated and imaged using finite element analysis (Figure 4c) [37]. The BTO with tetragonal crystal structure (T−BTO) features the strongest piezoelectric performance than other crystalline forms [38]. Thanks to its facile synthesis, good piezoelectric property as well as superior stability, BTO NMs have been widely used in piezocatalysis and piezo−assisted photocatalysis [39]. More importantly, BTO demonstrates good biocompatibility, making it a promising biomedical material. Numerous inorganic functional components have thus been integrated with BTO for tumor treatment, such as Au nanoparticles [40], CuO, etc. [41], further increasing the efficiency of the PEDT process.

(K, Na)NbO_3_ (KNN), a newly developed lead−free piezoelectric NM, offers better environmental and biological safety than the traditional lead−based Pb(Zr, Ti)O_3_ ones. Phase boundary engineering to KNN can further enhance the piezoelectric property. Upon the reduction in the piezoelectric domain size of KNN to nanoscale, the boundary energy is largely decreased, which increases the mobility of charges and subsequent piezo−response. Moreover, due to the high oxygen vacancy content in KNN lattice, the diamond and quadrangular KNNs are able to hasten the demise of cancer cells [39]. A recent study showed that piezoelectric materials, such as BTO, are not biodegradable and may even be toxic in living organisms. By contrast, lead−free KNN is not only less harmful, but also effectively biodegradable in living organisms without significant negative effects [27,42], thus rendering KNN NMs highly promising for biomedical applications. 

PVDF, a semi−crystalline polymer, is the most investigated organic piezoelectric material in bio−research. In a polymer chain of PVDF (Figure 5a), the presence of an electronegative fluorine and an electropositive hydrogen forms a net dipole moment. Figure 5b demonstrates that PVDF chains can form in parallel; therefore, in a ferroelectric state, all the chain dipoles are lined up along the crystallization axis, resulting in macroscopic polarization [43]. In comparison to other piezoelectric polymers, e.g., poly(vinylidene fluoride−trifluoroethylene) (P(VDF−TrFE)), PVDF possesses the highest piezoelectric coefficient and the lowest acoustic impedance. Baniasadi et al. measured the charges produced by the PVDF−TrFE nanofibers under mechanical deformation [44]. As illustrated in Figure 5c, the PVDF−TrFE nanofibers were placed in a periodically bending device with two electrodes linked to a multimeter, positioned on the flexure stage for the bending−induced piezoelectric potential (voltage) measurement (Figure 5d). The flexure stage produced two oppositely oriented peaks. The bending deformation caused the first peak, followed by the relaxation−state−induced second peak. The homing piezoelectricity was also examined using PFM, as shown in Figure 5e,f, where the piezoelectric response of a single nanofiber varies linearly with the applied voltage, with a measured piezoelectric constant of 37−48 pm/V, which is slightly higher than that of the P(VD−TrFE) (38 pm/V), indicating that the nanofibers made by electrostatic spinning possess stronger piezoelectric capabilities. Featuring high biocompatibility, corrosion resistance, thermal stability and ageing resistance, PVDF has been an ideal platform for wearable and implantable energy−converting applications [45,46]. With the strong flexibility and ductility, PVDF can be easily deformed when subjected to ultrasound, inducing an internal polarization to generate surface charges that can take part in redox reactions [47,48]. Nonetheless, PVDF alone has rarely been involved in PEDT applications, due to its weak piezoelectric response compared with the inorganic materials, which usually needs to be augmented through a complex pretreatment of polarization.

Recently, some newly developed piezocatalytic materials have also been introduced for biomedicine research, such as BiOX(X=Cl, Br), black phosphorus (BP), Bi_2_MoO_6_ (BMO) and MoS_2_ [49,50,51], demonstrating interesting ultrasound−triggered bioactivity for PEDT.

## 4. Application of Piezoelectric NMs in PEDT

In general, the piezoelectric NMs used for PEDT can be categorized into three groups: (i) simple material system, where mainly the classic piezoelectric materials, such as BTO, KNN, etc., mentioned above, are used for PEDT; (ii) modified material system, where the crystal structure or surface of conventional piezoelectric materials are engineered by doping, hybridization or defecting, to improve the piezoelectric charge separation; (iii) multifunctional material system, in which the materials, combined with both piezoelectric and other field−responsive properties, such as photon or radiation energy conversion, were recently developed. 

### 4.1. Simple Material System for PEDT

The classic piezoelectric materials mentioned above have been industrialized and broadly used in various applications for several decades; their fundamental information, such as microstructure, ferro/piezoelectric parameters as well as chemical properties, is therefore more accessible than other newly developed ones, rendering them cost−effective and reliable candidates in biomedicine research. 

With their favorable biocompatibility and stability, BTO ceramics are highlighted as a safe piezoelectric platform for tumor eradication. As shown in Figure 6, Shi et al [38] developed a hydrogel compound containing tetragonal crystalline BTO nanocubes (T−BTO−GEL) as medicine for PEDT. In principle, the bandgap structure of the T−BTO is energetically unfavorable for the formation of •OH and •O_2_^−^ radicals, due to its bandgap of 2.56 eV with the flat band potential of −0.16 V vs. the normal hydrogen electrode (NHE). In this work, with the external pressure of ultrasound, a built−in piezoelectric field was formed, inducing the band bending of BTO, which enables the surface reactions to form the ROS to eradicate 4T1−tumors in vivo. After the piezocatalytic therapeutic treatment, no obvious damage to the major organs (e.g., heart, liver, spleen, etc.) were observed, indicating an effective and safe process of PEDT.

BTO nanoparticles have also been used in the study of PEDT for triple−negative breast cancer. The piezoelectric field intensity of the BTO crystal structure of cubic (BTO), tetragonal unpolarized (U−BTO) and tetragonal (T−BTO) types were measured to be 0 mV, 425 mV and 886 mV, respectively. In vitro tests revealed that the piezoelectric electric stimulation of T−BTO effectively reduced the proliferative potential of cancer cells, inhibiting tumor cells from multiplying and metastasizing. Both in vitro and in vivo investigations indicated that the piezoelectric stimulation enhanced the hypoxia of the tumor microenvironment and generated ROS to damage the mitochondria of the breast cancer cells [52].

Defective lead−free piezoelectric KNN has been used for PEDT of osteosarcoma (OS). Significant anti−tumor effects on human OS cells were achieved upon the production of ROS, induced by the ultrasound−triggered piezoelectric catalytic process and further enhanced by the presence of oxygen vacancies in KNN. Moreover, the KNN mainly caused apoptosis and autophagy of the OS cells with good biocompatibility in vivo, without showing side effects on normal cells and organs [53]. The cytotoxicity tests of KNN in 143B and KHOS are shown in Figure 7a and Figure 7b, respectively, indicating that KNN significantly inhibits tumor growth when subjected to ultrasound. Figure 7c–h displays the tumor image from the 143B and KHOS cell−implanted xenograft; the results of tumor mass and growth curves indicate that the KNN−injected mice had significantly less tumor weight and limited tumor growth under ultrasonication. 

Two−dimensional (2D) Bi_2_MoO_6_ nanoribbons (BMO NRs) were synthesized as piezoelectric sonosensitizers for glutathione (GSH)−enhanced PEDT. Upon band bending induced by the piezoelectric effect, endogenous GSH can be consumed by BMO NRs to suppress redox homeostasis, and the GSH−activated BMO NRs with an oxygen−deficient structure can favor the electron−hole pair separation owing to the brief spikes of high piezoelectric potential, thus improving the ROS production in PEDT [49]. Piezoelectric black phosphorus (BP) nanosheets, as an acoustic sensitizer, can produce •OH under ultrasonic circumstances for PEDT. Free radicals were produced via the surface reaction of the oxygen molecules and surface electrons, which was sped up by piezoelectric−induced band bending under the strain of ultrasound [50]. BiOCl nanoparticles were used for the treatment of malignancies with hydrogen peroxide (H_2_O_2_) as the main product of the piezoelectric catalytic process. Under the assistance of X−ray radiation, the production of hydroxyl radicals was also enhanced, which can further accelerate tumor destruction with the increased ROS generation [51].

### 4.2. Modified Material System for PEDT

To increase the PEDT effect, it is crucial to enhance the separation of electron–hole pairs as well as suppress their recombination. Additional components have thus been introduced to the simple material system as dopants or sensitizers, which can tune the mobility of charge carriers, to generate ROS more efficiently. For instance, it is reported that defect engineering by integrating Gd into ZnO lattice can create surface oxygen vacancies (D−ZnOx−Gd), providing active sites to enhance the electron capture and reducing electron–hole complexation. The oxygen defect−rich materials showed an improved capacity as high as 0.15− or 0.25−fold to adsorb oxygen molecules and water from tumor cells, compared to those without defect engineering, favoring ROS production [54]. Additionally, D−ZnOx−Gd can absorb near−infrared (NIR) light, allowing an additional photothermal process to further kill cancer cells by converting photon energy into heat through laser irradiation. Wu et al. also showed that doping Bi onto flawed BTO tetrads led to the dual functions of effectively inhibiting the recombination of electrons and holes to improve ROS production and increasing the bio−compatibility of BTO. In addition, Bi is a cost−effective metallic element compared to other transition metals, e.g., Pt, for surface modification [55].

Calcium phosphate (Ca_3_(PO_4_)_2_) features good biocompatibility but has poor piezocatalytic efficiency. Nikhil and co−workers reported the decoration of Au nanoparticles on Ca_3_(PO_4_)_2_ nanoparticles to improve the piezocatalytic property by a factor of 10, with a piezoelectric constant value of 72 pm/V, approaching the performance of the classic BTO. The generation of ROS using Ca_3_(PO_4_)_2_−Au nanohybrids was significantly increased for efficient PEDT [56]. KNN was doped with Se, a trace element needed by the human body, to modify the charge mobility. Se increased the piezoelectric capabilities of the polarized KNN by increasing its surface potential and by controlling the electric domain. Moreover, Se can also support the immune and preventative functions of the human body to cancerous cells [57]. To increase the effectiveness of drug administration, a piezo−drug nanocarrier was developed by combining MoS_2_ and molecular medicine to lower the interstitial tumor plasma fluid pressure. Upon ultrasound stimulation, •OH radicals are produced in the presence of MoS_2_ via a piezocatalytic process, which can interact with doxorubicin to lower tumor cell plasma fluid pressure, thereby increasing drug release and slowing tumor growth [58].

### 4.3. Multifunctional Material System for PEDT

To gain satisfying therapeutic effects or to increase treatment efficiency, such as with a shorter duration time or wild external stimulation, the use of multifunctional NMs with multi−stimulating external fields is more efficient than with the piezocatalytic process alone. For instance, a nanohybrid system, by integrating the piezoelectric and NIR light absorption components into a single unit, can convert the absorbed photon energy to thermal energy or exited charge carriers. 

Piezoelectric–photothermal therapy using a nanohybrid with piezoelectric and photothermal properties is a non−invasive treatment method that combines PEDT and PTT to kill cancer cells. Usually, the PTT efficiency decays significantly with the increase in the thickness of the tissue around tumors; thus, PTT alone mainly works for superficial tumors, such as skin and esophageal cancers [59]. By endowing photothermal conversion material with piezoelectric activity, the resulted PEDT−PTT combination is highly effective for deep−seated malignancies.

The piezoelectric field can also promote the transfer of the photo−excited charge carriers and reduce the recombination of electrons and holes, thus increasing the efficiency of redox reactions for ROS production via a PEDT−PDT process. Sphalerite nanosheets, bearing both piezoelectricity and photoelectricity, have been used for cancer treatment. Under the stimulation of ultrasound and irradiation of light, the PEDT−PDT process produced concentrated ROS by consuming glutathione inside the tumor, creating an imbalance in the internal environment of the tumor and causing apoptosis. The ROS can also be generated solely at the tumor site with this method, facilitating the death of tumor cells [60].

In addition, a nanocomposite composed of Au and ZnO nanoparticles dispersed on graphene nanosheets has shown improved capability in cancer treatment. The graphene can transfer piezoelectric−effect−induced surface electrons rapidly, while the Au on graphene can trap the electrons and act as reaction center to form ROS [61]. The graphene demonstrated dual functions of promoting the overall PEDT effect, as the electron transport highway owing to its good conductivity, and as additional nanomedicine, due to its weakly acidic surface [61]. BTO−Cu_2−x_O nanohybrid was used as a sonosensitizer and CDT agent simultaneously for improving ROS generation and cancer therapy. Using ultrasound, the ^1^O_2_ and •OH radicals were produced via a piezocatalytic process to initiate the PEDT (Figure 8). Due to the presence of Cu_2−x_O on BTO’s surface, a Fenton−like reaction can also be triggered to convert endogenous H_2_O_2_ into •OH for CDT, again enhancing the overall therapeutic outcome by damaging cellular mitochondria of refractory breast cancer cells [41].

## 5. Outlook and Conclusions

As demonstrated above, the concept using piezoelectric NMs and ultrasound excitation to generate ROS have paved a promising route for cancer therapy. However, there are still significant challenges and opportunities that need to be systematically investigated to further develop PEDT. 

First, most of the piezoelectric NMs used for PEDT listed in this review only exhibit a weak piezoelectric response. In fact, there are plenty of piezo/ferroelectric materials with high piezoelectric properties; for instance, some transition metal carbide (TMC) NMs, e.g., SnSe, show superior piezocatalytic activity, considerably outperforming the conventional BTO and KNN [62]. Therefore, it is of high technical value to examine these new materials for PEDT application. It is noted that unlike the conventional BTO and ZnO, whose biosafety and biocompatibility have been widely acknowledged, the toxicity and biosafety of TMC are unknown, requiring a systematic study prior to in vivo application. 

Second, the influence of the piezoelectric nanoparticle morphology on the material–cell interaction could be optimized to enhance the overall therapeutic results. In general, the particles with size <100 nm are expected to be more favorable for endocytic cell uptake and the following dynamic treatment process intracellularly. Currently, the size of most reported NMs for PEDT are larger than 100 nm, which is not suitable for cellular uptake and intracellular reactions, whereas they might be active for extracellular treatment to the plasma membrane. Size selection/optimization based on tumor cell characteristics is suggested to increase the efficiency of dynamic therapy. Moreover, 1D or 2D nanostructures, such as nanosheets or nanowires, demonstrate stronger piezoelectric response than other morphologies, as they can be easily deformed under strain [62]. Therefore, the use of 1D or 2D piezoelectric nanostructures may result in better PEDT results. 

Third, to increase the ROS production efficiency, a rapid charge transfer from the surface of the piezoelectric NMs is critical. To this end, hybridizing them with other semiconductors to align the band structure at the junction or with conductive materials is effective and easily accessible manipulation. For example, the Z−scheme band hybrid structure, composed of two semiconductors, which has been widely studied in photocatalysis, could be introduced to the piezocatalytic system to promote the surface charge separation; moreover, decoration of the piezoelectric semiconductors with metal or carbon materials, such as carbon dots or carbon nitride, to increase their surface conductivity is also feasible to speed up surface charge mobilization [63]. The nanohybrids can be designed with multi−stimulation response by combining piezoelectric with other properties into one unit, such as response to radiation or magnetic field, to generate a synergistic effect, e.g., the coupling of PEDT with PTT or PEDT with PDT, eventually leading to better therapeutic results. 

Currently, most studies using piezoelectric biomaterials in tumor therapy are focused on solid tumor cell treatment; little research has shown their application in therapy for hematologic malignancies, which are highly challenging to cure, such as leukemia and lymphoma. Unlike solid tumors, where cancerous cells are concentrated in a confined region, hematologic malignancy cells are widely distributed throughout the body, bringing difficulty in nanomedicine targeting and ROS accumulation, which largely reduces the therapy’s effect. Therefore, the use of piezoelectric biomaterials for the treatment of hematologic malignancies requires more in−depth exploration.

Finally, like other functional bio−NM systems, despite the good bio−compatibility and promising application of piezoelectric NMs for tumor therapy, their long−term in vivo stability and toxicity are still under−investigated. In general, these materials cannot be degraded and easily eliminated/excreted from the body, which presents major obstacles to their clinical implementation. Although fundamental research shows that the piezoelectric materials are non−toxic to living organisms, their accumulation in the organs, which cannot be metabolized, may result in a generation excess of ROS, which may cause damage to normal cells.

In summary, using piezoelectric NMs to convert the mechanical energy of ultrasound stimulation to electric energy has afforded an effective and safe process of dynamic treatment to eliminate cancer cells. The investigation of piezoelectric biomaterial system and its PEDT application is still at the initial stage in the research domain of dynamic tumor therapy. In addition, targeted treatment after drug injection into the body and simultaneous treatment of multiple cancer cells still need further exploration. To gain more fundamental achievements and promote the clinical translation in the future, it is of great significance to pursue not only material innovations but also their in vivo examination.

## Figures and Tables

**Figure 1 molecules-28-03170-f001:**
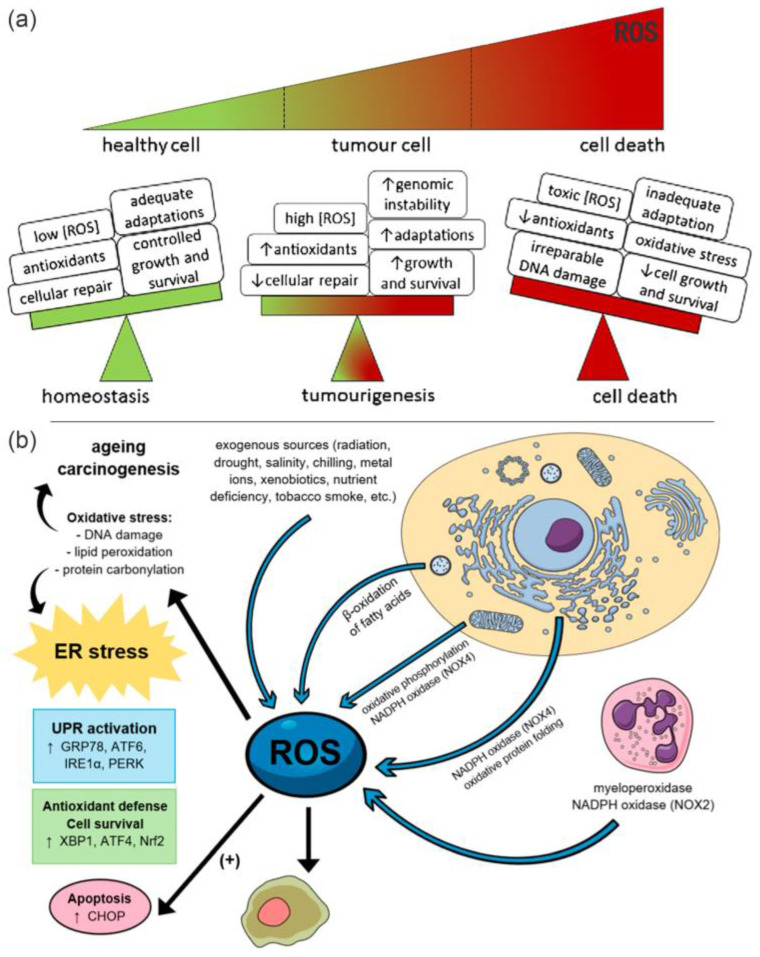
(**a**) Effect of ROS level on cells. Reproduced with permission [18]. Copyright 2018, Elsevier Ltd. (**b**) ROS−induced damage to the cells. The arrows indicate the sources of ROS and their effects on cells. Reproduced with permission [19]. Copyright 2019, MDPI (Basel, Switzerland). Healthy cells can keep adequate adaptations to overcome the damaging effects of ROS. Elevated ROS levels at toxic concentrations can generate oxidative stress, resulting in irreparable damage to the cell, inadequate adaptations and eventually tumor cell death.

**Figure 2 molecules-28-03170-f002:**
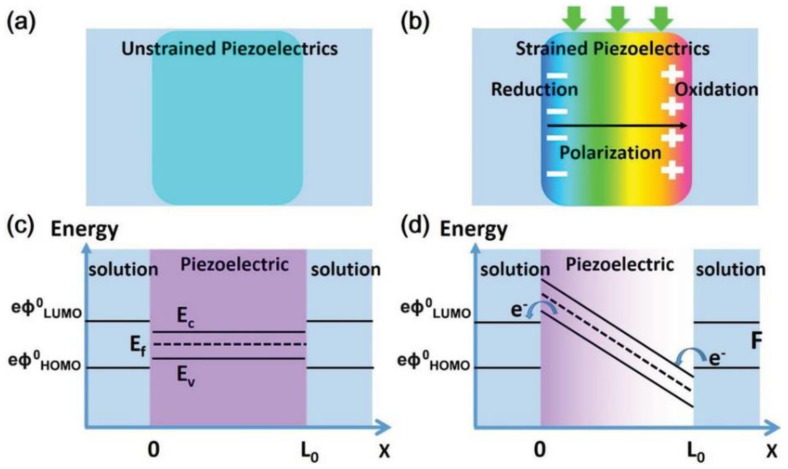
(**a**,**b**) Schematic diagrams show the formation of the piezoelectric field via band bending. (**c**) Band diagram of piezoelectric materials without (**c**) and with (**d**) strain force. Upon the band bending, the piezoelectric field changes the energy state across the material, allowing charge transfer at the material−solution interface. Reproduced with permission [20]. Copyright 2015, Elsevier Ltd.

**Figure 3 molecules-28-03170-f003:**
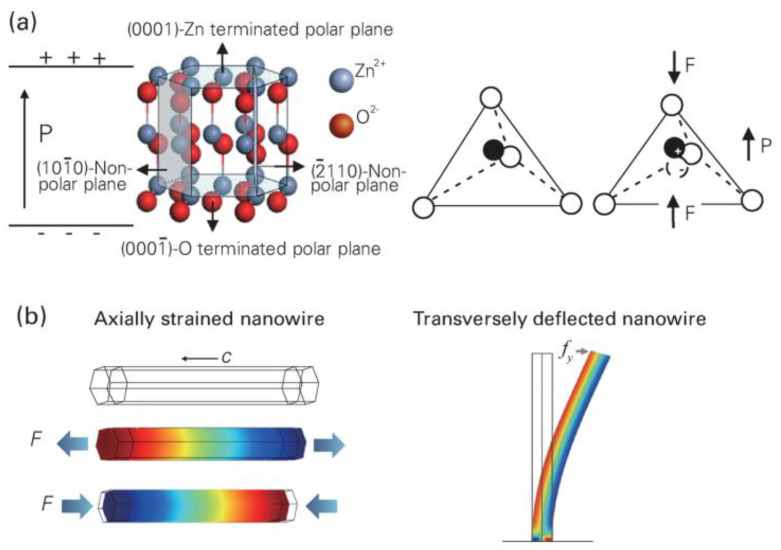
(**a**) Atomic model of the wurtzite−structured ZnO. (**b**) Finite element analysis simulated piezoelectric potential distribution in a ZnO nanowire. Three−dimensional graphs of the nanowire piezoelectric potential distribution and deformation shape at 85 nN of stretching force (middle) and 85 nN of compressing force (bottom). Reproduced with permission [30]. Copyright 2018, IOP Publishing.

**Figure 4 molecules-28-03170-f004:**
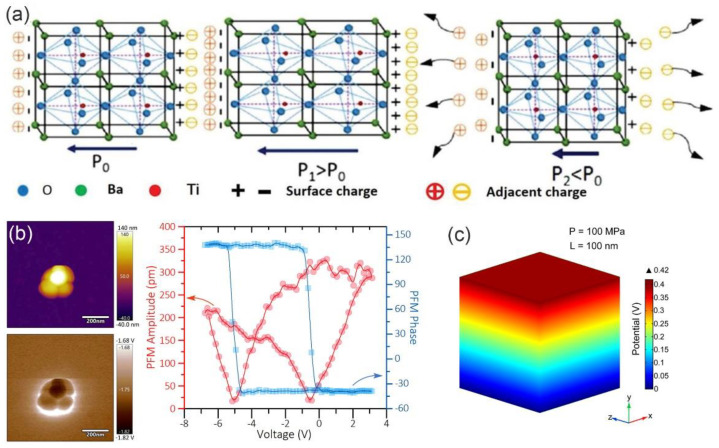
(**a**) Schematic illustration of charge migration within BTO unit cell with variations in stress. Reproduced with permission [36]. Copyright 2015, American Chemical Society. (**b**) Topographic image and corresponding phase hysteresis and amplitude butterfly loops of a BTO nanocube measured using PFM. (**c**) Finite element analysis of the electric field of BTO nanocubes at pressure of 100 MPa. Reproduced with permission [37]. Copyright 2022, American Chemical Society.

**Figure 5 molecules-28-03170-f005:**
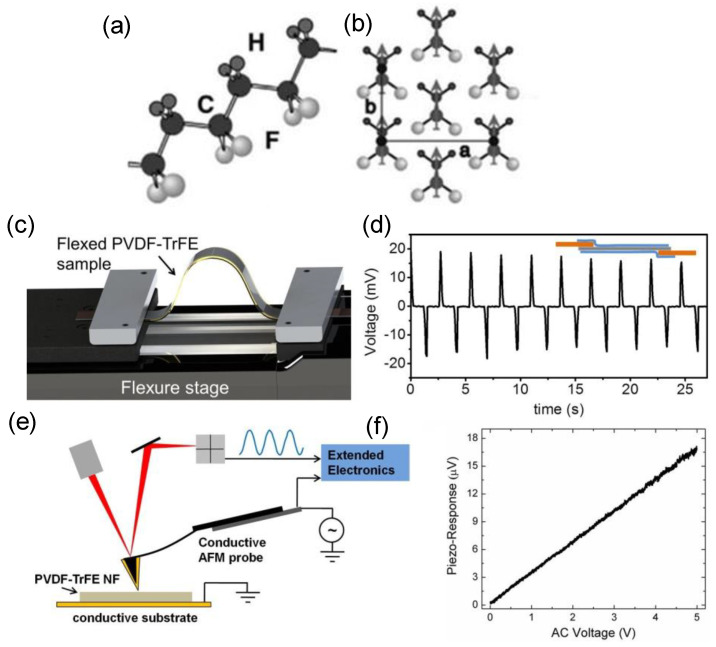
(**a**) Chain structure PVDF in the all−trans conformations, which exhibits the planar carbon backbone with coupled F and H atoms, and (**b**) the crystal structure of the phase. In this figure, a and b represent the parallel alignment of the junction axes and the direction of polarization, respectively. Reproduced with permission [43]. Copyright 2023, Royal Society of Chemistry. (**c**) Piezoelectric characterization of the electro−spun nanofibers. (**d**) Flexure test experiment. Plot depicts the generated voltage for a number of flexed and unflexed states; inset shows a schematic of the ready sample. (**e**) PFM experimental schematic of PVDF−TrFE nanofiber and (**f**) its piezo−response amplitude vs. applied voltage. Reproduced with permission [44]. Copyright 2015, American Chemical Society.

**Figure 6 molecules-28-03170-f006:**
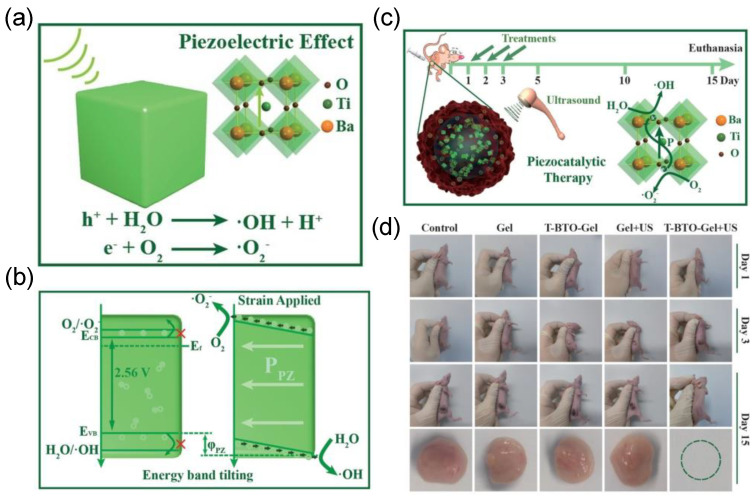
(**a**) Schematic diagram of •OH and •O_2_^−^ generated on T−BTO under ultrasonic conditions. (**b**) Energy band bending under ultrasonication. (**c**) Schematic diagram of piezoelectric therapy for tumor. (**d**) Tumor treatment results under different conditions. Reproduced with permission [38]. Copyright 2020, Wiley−VCH.

**Figure 7 molecules-28-03170-f007:**
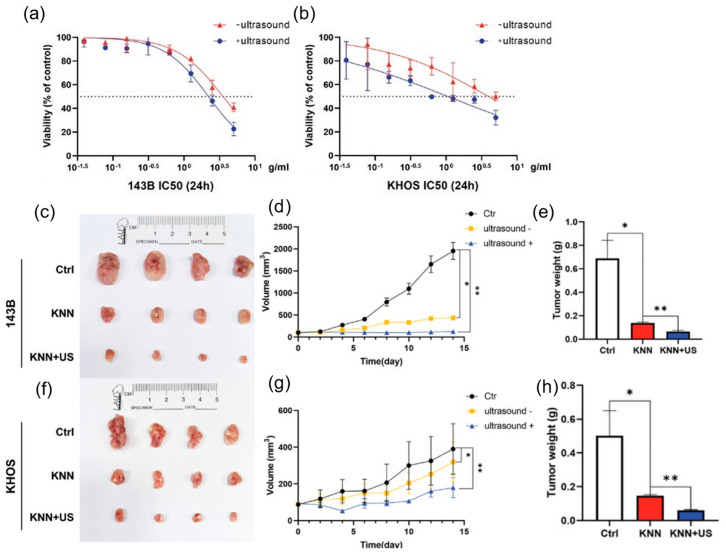
(**a**,**b**) 143B and KHOS cell viability in the presence of KNN with and without ultrasonication. (**c**) Photo images of the tumors from 143B cell xenografts under different treatments. (**d**,**e**) 143 B cell xenograft tumor growth (n = 4, * *p* < 0.05, ** *p* < 0.01). (**f**) Images of tumors created using KHOS cells from the xenografts of different treatments. (**g**,**h**) KHOS cell xenograft tumor growth. Reproduced with permission (n = 4, * *p* < 0.05, ** *p* < 0.01) [53]. Copyright 2022, Wiley−VCH.

**Figure 8 molecules-28-03170-f008:**
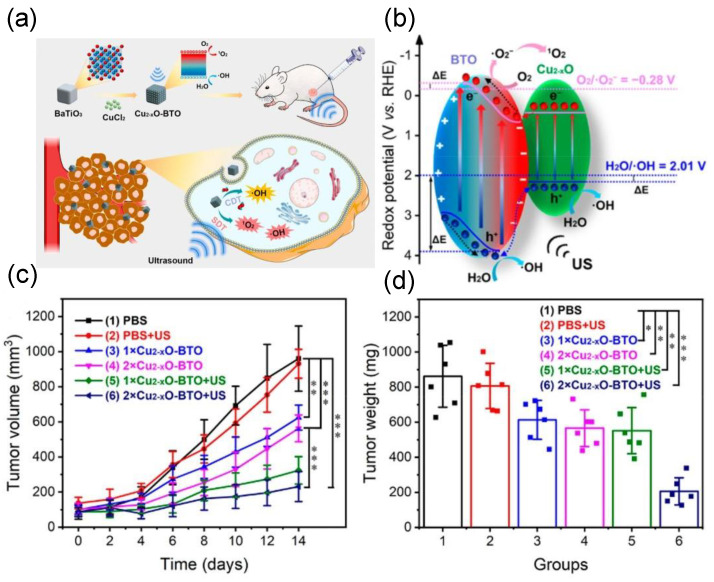
(**a**) Schematic diagram of the PEDT−CDT using BTO−Cu_2−x_O nanohybrid. (**b**) Mechanism of the ROS generation using BTO−Cu_2−x_O. (**c**,**d**) Average tumor growth curves of the mice and average tumor weight after the different treatments (*** *p* < 0.001, ** *p* < 0.01, * *p* < 0.05). Reproduced with permission [41]. Copyright 2022, American Chemical Society.

## Data Availability

Not applicable.

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
