# Peer review of "Emerging Advancements in Piezoelectric Nanomaterials for Dynamic Tumor Therapy"

_molecules, 2023, doi:10.3390/molecules28073170_

Round 1
Reviewer 1 Report
The present review manuscript entitled “Emerging Advances for Piezoelectric Nanomaterials based for Dynamic Tumor Therapy” authored by Qian Yu et al. describes the achievements of piezoelectric biomaterials for tumor therapy, including the mechanism and advances of the state-of-the-art tumor treatment application. It is a well-written review article and absence of major errors. Therefore, I endorse it for publication. However, certain Minor issues are detailed below which need to be addressed before its final acceptance in the Molecules.
Comment 1: Firstly, I would like to draw the attention of the authors that there are some typographical errors in the manuscript, so the authors need to correct them in the revised manuscript.
Comment 2: Minor punctuation revision is required in the manuscript. Improve the English language of the manuscript.
Comment 3: Revise the abstract section, it should clearly discuss the problem statement and the current study approach. So, please enhance the abstract section. It can improve the visualization of the paper’s materials for the readers in the different areas of study.
Comment 4: Figure 1 quality is not good.
Comment 5: Authors can give more briefing of cytotoxicity Piezoelectric which is a much more important aspect in cancer therapy
Comment 6: In the introduction line 28 is not clear. Numerous strategies have been 27 developed, including surgery, radiotherapy, and chemotherapy et al,.complete the sentence with reference
Comment 7: Authors have mentioned this as a short review please specify if this article is a short review or review article
Comment 8: What are the limitations of the inclusion and exclusion of data collected for the review article
Comment 9: All the references quoted in this study from China province have any specific reason.
Comment 10: The author's contribution is missing from the article
Comment 11: Suggested adding abbreviations in the revised manuscript.
Author Response
Dear reviewer,
We are highly grateful to you for your pertinent and valuable comments and suggestions. Accordingly, we have made corresponding modifications to our manuscript. The manuscript after required revisions addresses all of the concerns raised by them. The detailed response is included in the response letter. Page numbers refer to the revised version.We are herein re-submitting the manuscript electronically with the main changes highlighted in red for your review.Thank you for your suggestions again.
Yours sincerely,
Prof. Jianming Zhang

Reviewer 2 Report
Cancer is one of the deadliest diseases. Chemotherapy is one of the conventional treatment methods. However conventional chemotherapy usually introduces various side effects, such as cytotoxicity or multi-drug resistance. Targeted therapy based on nanomaterials has been emerging as a promising method to treat malignancies with low toxicity and minimal side effects. Piezoelectric biomaterials can convert mechanical energy to electrical energy that can subsequently cause sonodynamic effect to selectively kill tumor cells. In this manuscript, the mechanism and advances of the-state-of-the-art tumor treatment application of piezoelectric biomaterials have been reviewed. This is an interesting review in the field and is publishable. Please consider the minor comments below to improve the manuscript.
1) Hematologic malignancy is significantly different from solid tumor. The authors should consider to specify the type of the cancers and related challenges for specific cancer.
2) The authors should consider to discuss a bit the translational capability of these piezoelectric biomaterials as well as the current technical challenges in the field.
Author Response

(The authors gave the same response as above.)
